# Optimality Conditions for Approximate Solutions of Set Optimization Problems with the Minkowski Difference

**Yuhe Zhang**  **and Qilin Wang \***

College of Mathematics and Statistics, Chongqing Jiaotong University, Chongqing 400074, China;
622210150034@mails.cqjtu.edu.cn
\* Correspondence: 990020040623@cqjtu.edu.cn

**Abstract:** In this paper, we study the optimality conditions for set optimization problems with set criterion. Firstly, we establish a few important properties of the Minkowski difference for sets. Then, we introduce the generalized second-order lower radial epiderivative for a set-valued maps by Minkowski difference, and discuss some of its properties. Finally, by virtue of the generalized second-order lower radial epiderivatives and the generalized second-order radial epiderivatives, we establish the necessary optimality conditions and sufficient optimality conditions of approximate Benson proper efficient solutions and approximate weakly minimal solutions of unconstrained set optimization problems without convexity conditions, respectively. Some examples are provided to illustrate the main results obtained.

**Keywords:** set optimization problems; optimality conditions; generalized second-order lower radial epiderivatives; minkowski difference

**MSC:** 49Q46; 54C60; 90C26

## 1. Introduction

Set-valued optimization is a kind of extension of vector optimization, which has become a flourishing branch of applied mathematics due to the application of set-valued optimization problems in many fields [1–3]. It is widely known that the analysis of optimality conditions for various types of set optimization problems and their solutions strongly depends on the features of set-valued maps and their derivatives, or epiderivatives, see [4–6] . Based on a unique concept of the difference of sets, Jahn [7] presented the idea of the directional derivative of a set-valued map and used the derivative to derive the optimality conditions for a set optimization problems. In order to figure out the optimality conditions of the $\ell$-minimal solution for a set optimization problem, Durea and Strugariu [8] proposed the concept of the directional derivative of set-valued maps. Using the modified Demyanov difference and the derivative, Dempe and Pilecka [9] defined the directional derivative of the set-valued maps and developed the optimality condition for the set optimization problem. Since the radial set of a set contains global information of the set [10–12], radial derivatives [13] have drawn a lot of attention, see [14,15]. For constrained set-valued optimization problem, Yu [14] proposed the the higher-order radial derivative of set-valued maps, by means of the derivative, they developed optimality conditions for lower weak minimal solution.

In practical application, the mathematical programming models are usually not accurate enough, so the solutions of the model are generally approximate rather than exact. Meanwhile, approximate solutions can approximate exact solutions for mathematical programming problems. It is important to note that most of the real-world optimization issues, including economic analysis and traffic optimization, ecological planning, etc., have approximate solutions, which are highly helpful in the analysis and treatment of set-valued

optimization problems; see [16–18] for details. Therefore, the approximate solution of optimization problems has attracted much attention from many scholars, see [19,20]. The idea of $\varepsilon$-quasi solutions to vector optimization problems was first suggested by Loridan [21]. For a set-valued optimization problem, by combining vector and set criteria, Dhingra and Lalitha [22] introduced concepts of approximative solutions. In a locally convex Hausdorff topological vector space, Hu et al. [20] proposed an approximative Benson proper effective solution to the set-valued equilibrium problem and explored the relationship between the Benson effective solution and the approximate one. The Painlevé-Kuratowski lower and upper convergence of the approximation solution for set optimization problems under continuity and convexity are derived by Han et al. [23] . To gain the sufficient conditions of minimal solution sets, Gupta and Srivastava [24] introduced a novel concept of approximation weak minimal solution for a set optimization problem. Without employing the convexity, Han [25] obtained two scalarization theorems for the connectedness of the weak $l$-minimal approximate solutions for set optimization problems.

To our knowledge, there is relatively little literature on the second-order optimal conditions for approximate solutions of set optimization problems with the Minkowski difference. Motivated by the the derivatives in [14,15] and approximate solutions in [20,24], we introduce the generalized second-order lower radial epiderivative for set-valued maps, approximate Benson proper effective solutions and approximate weakly minimal solutions of the set optimization problem based on the Minkowski difference. By the second-order lower radial epiderivative, we establish the necessary optimality conditions and sufficient optimality conditions of approximate Benson solutions and approximate weakly minimal solutions for unconstrained set optimization problems, respectively.

The article is organized as follows. We recall some preliminaries and establish a few features of the Minkowski difference for sets in Section 2. We firstly propose the generalized second-order lower radial epiderivative for set-valued maps and discuss some properties of the epiderivative in Section 3. We discuss the second-order necessary and sufficient conditions for approximate Benson proper efficient solutions and approximate weakly minimal solutions of the unconstrained set optimization problems in Section 4. The brief conclusion of the paper is given in Section 5.

## 2. Preliminaries and Definitions

Throughout the paper, unless otherwise specified, let $V$ and $P$ be two normed spaces, $P^*$ be the topological dual space of $P$ , $\mathcal{P}(V)$ be the family of all nonempty subsets of $V$. We denote by $\text{int}(A)$ and $\text{cl}(A)$, the interior and closure of a set $A \subseteq P$, respectively. The generated cone of $A$ is defined by cone $A = \{ty \mid y \in A, t > 0\}$. In the sequel, $C$ is a solid $(\text{int}(C) \neq \varnothing)$ pointed $(C \cap (-C) = \{0\})$ closed convex cone in $P$. We have $C + C = C$, $C + \text{int}(C) = \text{int}(C)$ and $\lambda C = C$ for all $\lambda > 0$. The dual cone of $C$ is $C^* := \{b \in P^* \mid \langle b, z \rangle \geq 0, \forall z \in C\}$.

Let $S : V \rightrightarrows P$ be a set-valued map. The domain, graph, epigraph and profile map of $S$ are defined, respectively, by

$$\text{dom } S = \{v \in V \mid S(v) \neq \varnothing\},$$
$$\text{graph } S = \{(v, p) \in V \times P \mid p \in S(v)\},$$
$$\text{epi } S = \{(v, p) \in V \times P \mid p \in S(v) + C\},$$
$$S_+(v) = S(v) + C, v \in \text{dom } S.$$

Clearly, epi $S = $ graph $S_+$.

**Definition 1** ([26]). *Let $I, W \in \mathcal{P}(P)$. The Minkowski difference of $I$ and $W$ is defined as*

$$I \dot{-} W = \{p \in P \mid p + W \subseteq I\} = \bigcap_{w \in W} (I - \{w\}).$$

By the definition of Minkowski difference, the following results obviously hold.

**Proposition 1.** *Let $I, W, A \in \mathcal{P}(P)$. Then*
　*(a) $y \in I \dot{-} W$ if and only if $y \in I - \{w\}, \forall w \in W$.*
　*(b) $I \subseteq W \Rightarrow I \dot{-} A \subseteq W \dot{-} A$.*
　*(c) $I \subseteq W \Rightarrow A \dot{-} W \subseteq A \dot{-} I$.*
　*(d) $(I \bigcup W) \dot{-} A = (I \dot{-} A) \bigcup (W \dot{-} A)$.*

**Proposition 2.** *Let $I, W \in \mathcal{P}(P)$, $a, b \in P$, $M := \{a\} \times I$ and $N := \{b\} \times W$. Then*

$$u \in I \dot{-} W \Leftrightarrow (a - b, u) \in M \dot{-} N.$$

**Proof.** (i) ($\Rightarrow$) Let $u \in I \dot{-} W$. Then, it follows from Proposition 1 (*a*) that

$$u \in I - \{w\}, \forall w \in W.$$

Therefore,

$$(a - b, u) \in \{a - b\} \times (I - \{w\}) = (\{a\} \times I) - \{(b, w)\}, \forall w \in W,$$

that is,

$$(a - b, u) \in M - \{v\}, \forall v \in N,$$

which implies

$$(a - b, u) \in M \dot{-} N.$$

(ii) ($\Leftarrow$) Let $(a - b, u) \in M \dot{-} N$. Then, from Proposition 1 (*a*), we get

$$(a - b, u) \in M - \{v\}, \forall v \in N.$$

Therefore,

$$(a - b, u) \in \{a - b\} \times (I - \{w\}), \forall w \in W,$$

then

$$u \in I - \{w\}, \forall w \in W,$$

which implies

$$u \in M \dot{-} N.$$

The proof is complete.　□

**Lemma 1** ([26]). *Let $I \in \mathcal{P}(P)$. If $I$ is a convex set, then for any $W \in \mathcal{P}(P)$, the Minkowski difference $I \dot{-} W$ is a convex set.*

**Lemma 2** ([27]). *If $I, W \in \mathcal{P}(P)$ and $a \in P$, then*
　*(a) $(a + I) \dot{-} W = a + (I \dot{-} W)$.*
　*(b) $I \dot{-} (a + W) = -a + (I \dot{-} W)$.*
　*(c) If $I$ is closed, then $I \dot{-} W$ is also closed.*

**Definition 2** ([15]). *Let $G, H \in \mathcal{P}(V)$ and $t \in V$.*
　*(i)The generalized radial set of $G$ on $H$ is defined by*

$$T_r(G, H) := \{q \in V \mid \exists m_n > 0, \exists q_n \to q, s.t. \forall v_0 \in H, \forall n \in \mathbb{N}, v_0 + m_n q_n \in G\}.$$

　*(ii) The generalized second-order radial set of $G$ on $H$ with respect to $t$ is defined by*

$$T_r''(G, H, t) := \{q \in V \mid \exists m_n > 0, \exists q_n \to q, s.t. \forall v_0 \in H, \forall n \in \mathbb{N}, v_0 + m_n t + m_n^2 q_n \in G\}.$$

Inspired by the mth-order lower radial set in [11] and the generalized second-order radial set in [15], we propose the notion of the generalized lower radial set and the generalized second-order lower radial set.

**Definition 3.** *Let $G, H \in \mathcal{P}(V)$ and $t \in V$ .*
*(i)The generalized lower radial set of $G$ on $H$ is defined by*

$$T_{l-r}(G,H) := \{q \in V \mid \forall m_n > 0, \exists q_n \to q, s.t. m_n q_n \in G \dot{-} H, \forall n \in \mathbb{N}\}.$$

*(ii) The generalized second-order lower radial set of $G$ on $H$ with respect to $t$ is defined by*

$$T''_{l-r}(G,H,t) := \left\{q \in V \mid \forall m_n > 0, \exists q_n \to q, s.t. \ m_n t + m_n^2 q_n \in G \dot{-} H, \forall n \in \mathbb{N}\right\}.$$

**Remark 1.** *If $G$ is convex, then $T''_{l-r}(G,H,t)$ is convex.*

**Remark 2.** *$T_r(G,H) = T''_r(G,H,t) = T''_{l-r}(G,H,t) = \varnothing$ if and only if $G \dot{-} H = \varnothing$.*

**Remark 3.** *If the set $H$ is a singleton and $H \subseteq G$, then the generalized radial set $T_r(G,H)$ reduces to the closed radial cone $R(G, v_0)$ introduced in [13], the generalized second-order radial set $T''_r(G,H,t)$ reduces to second-order upper radial set $T_S^{r(2)}(v_0, t_0)$ introduced in [11] and the generalized second-order lower radial set $T''_{l-r}(G,H,t)$ reduces to second-order lower radial set $T_S^{r\flat(2)}(v_0, t_0)$ introduced in [11].*

**Remark 4 ([15]).** *Let $G, H \in \mathcal{P}(V)$ be two nonempty sets.*
*(i) $T_r(G,H) = \mathrm{clcone}(G \dot{-} H)$.*
*(ii) $T''_r(G,H,z) = \mathrm{cl} \cup_{t>0} \frac{G \dot{-} H - mz}{m^2}$.*
*(iii) If $G \dot{-} H \neq \varnothing$, then $T_r(G,H)$ is a nonempty closed cone, $T''_r(G,H,t)$ is a nonempty closed set such that $0 \in T''_r(G,H,t)$. $T''_r(G,H,t)$ is not a cone in general.*

By Definitions 2 and 3, the following results naturally hold.

**Proposition 3.** *Let $G, H, I \in \mathcal{P}(V)$ be three nonempty sets and let $v_0 \in G$. Then*
*(i) $T_{l-r}(G, v_0) \subseteq T_r(G, v_0)$.*
*(ii) $G \subseteq H \Longrightarrow T_{l-r}(G, v_0) \subseteq T_{l-r}(H, v_0)$.*
*(iii) $G \subseteq H \Longrightarrow T_{l-r}(G, I) \subseteq T_{l-r}(H, I)$ and $T_{l-r}(I, H) \subseteq T_{l-r}(I, G)$.*

Note that the inverse inclusion of Proposition 3 $(i)$ may not hold by the following example.

**Example 1.** *Let $V = \mathbb{R}^2$, $G = \left\{(p_1, p_2) \in \mathbb{R}^2 \mid p_1 \geq 0, p_2 \geq \frac{p_1}{2}\right\}$ and $H = \left\{(a, \frac{a}{2}) \in \mathbb{R}^2\right\}$. By calculating, we obtain*

$$T_r(G, v_0) = \left\{(v_1, v_2) \in \mathbb{R}^2 \mid v_1 \geq 0, v_2 \geq v_1\right\} \cup \left\{(v_1, v_2) \in \mathbb{R}^2 \mid v_1 \leq 0, v_2 \geq 0\right\}$$

*and*

$$T_{l-r}(G, v_0) = \left\{(v_1, v_2) \in \mathbb{R}^2 \mid v_1 \geq 0, v_2 \geq v_1\right\}.$$

*Thus, $T_r(G, v_0) \nsubseteq T_{l-r}(G, v_0)$.*

## 3. Generalized Second-Order Lower Radial Epiderivatives for Set-Valued Maps

In this section, by virtue of the Minkowski difference, we introduce generalized second-order lower radial epiderivatives for set-valued maps, and then investigate some characteristics of the epiderivative and generalized second-order radial epiderivatives. Firstly, we recall two concepts in [15].

**Definition 4** ([15]). *Let $S : V \rightrightarrows P$ be a set-valued map, $v_0 \in$ dom $S$ and $(t_0, q_0) \in V \times P$.*

*(i) The generalized radial derivatives of $S$ at $v_0$ is the set-valued map $D_r S(v_0) : V \rightrightarrows P$ defined by*

$$D_r S(v_0)(v) := \{p \in P \mid (v, p) \in T_r(\text{graph } S, \{v_0\} \times S(v_0))\}, \forall v \in V.$$

*(ii) The generalized second-order radial derivatives of $S$ at $v_0$ with respect to $(t_0, q_0)$ is the set-valued map $D_r'' S(v_0, t_0, q_0) : V \rightrightarrows P$ defined by*

$$D_r'' S(v_0, t_0, q_0)(v) := \left\{ p \in Y \mid (v, p) \in T_r''(\text{graph } S, \{v_0\} \times S(v_0), (t_0, q_0)) \right\}, \forall v \in V.$$

Next, we introduce the generalized lower radial epiderivative and the generalized second-order lower radial epiderivative of a set-valued map.

**Definition 5.** *Let $S : V \rightrightarrows P$ be a set-valued map, $v_0 \in$ dom $S$ and $(t_0, q_0) \in V \times P$.*

*(i) The generalized lower radial epiderivatives of $S$ at $v_0$ is the set-valued map $D_{l-r}S_+(v_0) : V \rightrightarrows P$ defined by*

$$D_{l-r}S_+(v_0)(v) := \{p \in P \mid (v, p) \in T_{l-r}(\text{epi } S, \{v_0\} \times S(v_0)\}, \forall v \in V.$$

*(ii) The generalized second-order lower radial derivatives of $S$ at $v_0$ with respect to $(t_0, q_0)$ is the set-valued map $D_{l-r}'' S(v_0, t_0, q_0) : V \rightrightarrows P$ defined by*

$$D_{l-r}'' S(v_0, t_0, q_0)(v) := \left\{ p \in P \mid (v, p) \in T_{l-r}''(\text{graph } S, \{v_0\} \times S(v_0), (t_0, q_0)) \right\}, \forall v \in V.$$

*(iii) The generalized second-order lower radial epiderivatives of $S$ at $v_0$ with respect to $(t_0, q_0)$ is the set-valued map $D_{l-r}'' S_+(v_0, t_0, q_0) : V \rightrightarrows P$ defined by*

$$D_{l-r}'' S_+(v_0, t_0, q_0)(v) := \left\{ p \in P \mid (v, p) \in T_{l-r}''(\text{epi } S, \{v_0\} \times S(v_0), (t_0, q_0)) \right\}, \forall v \in V.$$

**Remark 5.** *If $S(v_0) = \{p_0\}$, then $D_{l-r}'' S(v_0, t_0, q_0)$ reduces to the mth-order lower radial derivative $D_R^{\flat(2)} S(v_0, p_0, t_0, q_0)$ introduced in [13].*

**Proposition 4.** *Let $S : V \rightrightarrows P$ be a set-valued map, $v_0 \in$ dom $S$. Then $D_{l-r}S_+(v_0)$ is strictly positive homogeneous, i.e.,*

$$D_{l-r}S_+(v_0)(\alpha v) = \alpha D_{l-r}S_+(v_0)(v), \forall \alpha > 0.$$

**Proof.** Let $v \in V, \alpha > 0$.

(i) We first prove that

$$D_{l-r}S_+(v_0)(\alpha v) \subseteq \alpha D_{l-r}S_+(v_0)(v).$$

Let $p \in D_{l-r}S_+(v_0)(\alpha v)$. Then

$$(\alpha v, p) \in T_{l-r}(\text{epi } S, \{v_0\} \times S(v_0)).$$

Thus, for any sequence $\{m_n\}$ with $m_n > 0$, there exists a sequence $\{(v_n, p_n)\}$ with $(v_n, p_n) \to (\alpha v, p)$ such that

$$m_n(v_n, p_n) \in \text{epi } S \dot{-} \{v_0\} \times S(v_0).$$

Then

$$\alpha v m_n \left( \frac{1}{\alpha v} v_n, \frac{1}{\alpha v} p_n \right) \in \text{epi } S \dot{-} \{v_0\} \times S(v_0). \tag{1}$$

Naturally, $(\frac{1}{\alpha}v_n, \frac{1}{\alpha}p_n) \to (v, \frac{1}{\alpha}p)$. It follows from (1) that $\frac{1}{\alpha}p \in D_{l-r}S_+(v_0)(v)$. Therefore,

$$p \in \alpha D_{l-r}S_+(v_0)(v).$$

In this way,

$$D_{l-r}S_+(v_0)(\alpha v) \subseteq \alpha D_{l-r}S_+(v_0)(v).$$

(ii) Next, we prove that $\alpha D_{l-r}S_+(v_0)(v) \subseteq D_{l-r}S_+(v_0)(\alpha v)$.

The relationship of $\alpha D_{l-r}S_+(v_0)(v) \subseteq D_{l-r}S_+(v_0)(\alpha v)$ can be proved according to the same proof idea as (i).

So $D_{l-r}S_+(v_0)$ is strictly positive homogeneous. This completes the proof. $\square$

**Remark 6.** *It is clear that*

$$D''_{l-r}S_+(v_0, t_0, q_0)(v) \subseteq D''_r S_+(v_0, t_0, q_0)(v). \tag{2}$$

*However, the converse inclusions of* (2) *may not hold. The following example show the case.*

**Example 2.** *Consider set optimization problem with $V = P = \mathbb{R}$, $G = \{0, 1\}$ and $H = [1, +\infty)$. Let*

$$S(v) = \begin{cases} (-\infty, 2], v = 0, \\ (-\infty, 4], v = 1. \end{cases}$$

*It is obvious to get that* epi $S = G \times \mathbb{R}$. *So, for every $v_0 \in G$ and $(t_0, q_0) \in V \times (-H)$, we calculate that*

$$D''_r S_+(v_0, t_0, q_0)(x) = \mathbb{R}$$

*and*

$$D''_{l-r}S_+(v_0, t_0, q_0)(v) = [0, +\infty).$$

*Thus, $D''_r S_+(v_0, t_0, q_0)(v) \not\subseteq D''_{l-r}S(v_0, t_0, q_0)(v)$.*

**Remark 7.** *By Definitions 4 and 5, we get*

$$D_r S(v_0)(v) = D''_r S(x_0, 0, 0)(v)$$

*and*

$$D_{l-r}S_+(v_0)(v) = D''_{l-r}S_+(v_0, 0, 0)(x).$$

**Proposition 5.** *Let $S : V \rightrightarrows P$ be a set-valued map, $v_0 \in$ dom $S$ and $(t_0, q_0) \in V \times P$. Then*

$$D''_{l-r}S_+(v_0, t_0, q_0)(v) + C = D''_{l-r}S_+(v_0, t_0, q_0)(v), \forall v \in V.$$

**Proof.** (1) We first prove that

$$D''_{l-r}S_+(v_0, t_0, q_0)(v) + C \subseteq D''_{l-r}S_+(v_0, t_0, q_0)(x), \forall v \in V.$$

Let $v \in$ dom $D''_{l-r}S_+(v_0, t_0, q_0)$, $\bar{p} \in D''_{l-r}S_+(v_0, t_0, q_0)(v) + C$. Then there exist $c \in C$ and $p \in D''_{l-r}S_+(v_0, t_0, q_0)(v)$ such that

$$\bar{p} = p + c.$$

Since $p \in D''_{l-r}S_+(v_0, t_0, q_0)(v)$, for any sequence $\{m_n\}$ with $m_n > 0$, there exists a sequence $\{(v_n, p_n)\} \subset V \times P$ with $(v_n, p_n) \to (v, p)$ such that

$$\left(v_0 + m_n t_0 + m_n^2 v_n, p_0 + m_n q_0 + m_n^2 p_n\right) \in \text{epi } S, \forall p_0 \in S(v_0), n \in \mathbb{N},$$

that is,

$$p_0 + m_n q_0 + m_n^2 p_n \in S_+ \left( v_0 + m_n t_0 + m_n^2 v_n \right) + C, \forall p_0 \in S(v_0), n \in \mathbb{N}.$$

Set $\bar{p}_n := p_n + c$. Then $\bar{p}_n \to p + c$ and

$$
\begin{aligned}
p_0 + m_n q_0 + m_n^2 \bar{p}_n &= p_0 + m_n q_0 + m_n^2 (p_n + c) \\
&= p_0 + m_n q_0 + m_n^2 p_n + m_n^2 c \\
&\in S_+ \left( v_0 + m_n t_0 + m_n{}^2 q_n \right) + C,
\end{aligned}
$$

which implies that $p + c \in D''_{l-r} S_+(v_0, t_0, q_0)(v)$. Hence,

$$D''_{l-r} S_+(v_0, t_0, q_0)(v) + C \subseteq D''_{l-r} S_+(v_0, t_0, q_0)(v), \forall v \in V. \tag{3}$$

(2) We now prove that $D''_{l-r} S_+(v_0, t_0, q_0)(v) \subseteq D''_{l-r} S_+(v_0, t_0, q_0)(v) + C, \forall v \in V$. Since $0 \in C$, one gets

$$D''_{l-r} S_+(v_0, t_0, q_0)(v) \subseteq D''_{l-r} S_+(v_0, t_0, q_0)(v) + C, \forall v \in V. \tag{4}$$

From (3) and (4), we have

$$D''_{l-r} S_+(v_0, t_0, q_0)(v) + C = D''_{l-r} S_+(v_0, t_0, q_0)(v), \forall v \in V.$$

This proof is complete. □

**Proposition 6.** *Let $S : V \rightrightarrows P$ be a set-valued map, $v_0 \in \mathrm{dom}\ S$ and $(t_0, q_0) \in V \times P$. Then*

$$D''_{l-r} S(v_0, t_0, q_0)(v) + C \subseteq D''_{l-r} S_+(v_0, t_0, q_0)(v), \forall v \in V.$$

**Proof.** Let $v \in \mathrm{dom}\ D''_{l-r} S(v_0, t_0, q_0)$, $p \in D''_{l-r} S(v_0, t_0, q_0)(v), c \in C$. Then, for any sequence $\{m_n\}$ with $m_n > 0$, there exists a sequence $\{(v_n, p_n)\} \subset V \times P$ with $(v_n, p_n) \to (v, p)$ such that

$$\left( v_0 + m_n t_0 + m_n^2 v_n, p_0 + m_n q_0 + m_n^2 p_n \right) \in \mathrm{graph}\ S, \forall p_0 \in S(v_0), n \in \mathbb{N},$$

that is,

$$p_0 + m_n q_0 + m_n^2 p_n \in S \left( v_0 + m_n t_0 + m_n^2 v_n \right), \forall p_0 \in S(v_0), n \in \mathbb{N}.$$

Then

$$P_0 + m_n q_0 + m_n^2 (P_n + c) \in S \left( v_0 + m_n t_0 + m_n^2 v_n \right) + C, \forall p_0 \in S(v_0), n \in \mathbb{N}.$$

Set $\bar{p}_n := p_n + c$. Then $\bar{p}_n \to p + c$. So

$$p + c \in D''_{l-r} S_+(v_0, t_0, q_0)(v).$$

Hence,

$$D''_{l-r} S(v_0, t_0, q_0)(v) + C \subseteq D''_{l-r} S_+(v_0, t_0, q_0)(v), \forall v \in V. \tag{5}$$

This completes the proof. □

**Proposition 7.** *Let $E \subseteq V$ be a nonempty subset, $S : E \rightrightarrows P$ be a set-valued map and $(t_0, q_0) \in V \times P$. Then*

$$S(v) \dot{-} S(v_0) - \{q_0\} \subseteq D''_r S(v_0, t_0, q_0)(v - v_0 - t_0), \quad \forall v \in E.$$

**Proof.** It follows from Remark 4 (*ii*) that

$$T_r''(\text{graph } S, \{v_0\} \times S(v_0), (t_0, q_0)) = \text{cl} \cup_{m>0} \frac{\text{graph } S \dot{-} \{v_0\} \times S(v_0) - m(t_0, q_0)}{m^2} \tag{6}$$

Let $v \in E$ and $a \in S(v) \dot{-} S(v_0)$. Then it follows from Proposition 2 that for any $v \in E$, we get

$$(v - v_0, a) \in \{v\} \times S(v) \dot{-} \{v_0\} \times S(v_0)).$$

Therefore,

$$(v - v_0 - t_0, a - q_0) \in \{v\} \times S(v) \dot{-} \{v_0\} \times S(v_0) - (t_0, q_0))$$
$$\subset \text{cl} \cup_{m>0} \frac{\text{graph } S \dot{-} \{v_0\} \times S(v_0) - m(t_0, q_0)}{m^2}.$$

In combination with (6), we have

$$(v - v_0 - t_0, a - q_0) \in T_r''(\text{graph } S, \{v_0\} \times S(v_0), (t_0, q_0),$$

which implies that

$$a - q_0 \in D_r'' S(v_0, t_0, q_0)(v - v_0 - t_0).$$

Hence,

$$S(v) \dot{-} S(v_0) - \{q_0\} \subseteq D_r'' S(v_0, t_0, q_0)(v - v_0 - t_0), \quad \forall v \in E.$$

This completes the proof. $\square$

**Remark 8.** *Proposition 7 is established without any assumption of convexity.*

**Proposition 8.** *Let $E \subseteq V$ be a nonempty subset, $S : E \rightrightarrows P$ be a set-valued map and $(t_0, q_0) \in \{0\} \times C$. Then*

$$S(v) \dot{-} S(v_0) \subseteq D_r'' S(v_0, t_0, q_0)(v - v_0) + C.$$

**Proof.** From Proposition 7, we derive

$$S(v) \dot{-} S(v_0) - \{q_0\} \subseteq D_r'' S(v_0, t_0, q_0)(v - v_0 - t_0), \quad \forall v \in E.$$

Since $(t_0, q_0) \in \{0\} \times C$, for any $v \in E$, one has

$$S(v) \dot{-} S(v_0) \subseteq D_r'' S(v_0, t_0, q_0)(v - v_0) + C.$$

This completes the proof. $\square$

## 4. Optimality Conditions for Approximate Solutions of Set Optimization Problems

In this section, we discuss optimality conditions of approximate Benson proper efficient solution and approximate weakly minimal solutions for unconstrained set optimization problems by using the generalized second-order radial derivatives and the generalized second-order lower radial epiderivatives.

Let $S : V \rightrightarrows P$ be a set-valued map, $E \subseteq V$, we consider a set optimization problem $(\mathcal{SOP})$ as follows:

$$(\mathcal{SOP}) \begin{cases} \text{minimize } S(v) \\ \text{subject to } v \in E. \end{cases}$$

Next, we consider the following definitions for set optimization problem $(\mathcal{SOP})$ with the Minkowski difference.

**Definition 6** ([24])**.** *Let $\varepsilon > 0$ and $e \in \text{int}(C)$. A vector $v_0 \in E$ is said to be a $(\varepsilon, e)$-weak minimal solution of $(\mathcal{SOP})$, denoted by $v_0 \in (\varepsilon, e)$-WMin$(S, E, C)$, if*

$$(S(E) \overset{.}{-} S(v_0) + \varepsilon e) \cap (-\text{int}(C)) = \varnothing.$$

**Remark 9.** *(i) If $\varepsilon = 0$, then $(\varepsilon, e)$-weak minimal solution reduces to m-weak minimal solution considered in [24] for $(\mathcal{SOP})$.*

*(ii) If S is single-valued, then Definition 6 of $(\varepsilon, e)$-weak minimal solution reduces to the weak $\varepsilon e$-efficient solution for the vector optimization problems introduced in [28].*

Inspired by the Definition 6, we define MBenson proper efficient solution and $(\varepsilon, e)$-MBenson proper efficient solution with the Minkowski difference.

**Definition 7.** *Let $v_0 \in E$, $p_0 \in S(v_0)$, $\varepsilon \geq 0$ and $e \in \text{int}(C)$.*
*(i) $v_0$ is said to be a MBenson proper efficient solution of $(\mathcal{SOP})$, denoted by $v_0 \in MBenson(S, E, C)$, if*

$$clcone(S(E) \overset{.}{-} S(v_0) + C) \cap (-C \setminus \{0\}) = \varnothing.$$

*(ii) $v_0$ is said to be a $(\varepsilon, e)$-MBenson proper efficient solution of $(\mathcal{SOP})$, denoted by $v_0 \in (\varepsilon, e)$-MBenson$(S, E, C)$, if*

$$clcone\ (S(E) \overset{.}{-} S(v_0) + C + \varepsilon e) \cap (-C \setminus \{0\}) = \varnothing.$$

**Remark 10.** *(i) If $\varepsilon = 0$, then $(\varepsilon, e)$-MBenson proper efficient solution reduces to MBenson proper efficient solution for $(\mathcal{SOP})$.*
*(ii) For every $\varepsilon \geq 0$, we have $MBenson(S, E, C) \subseteq (\varepsilon, e)$-MBenson$(S, E, C)$.*
*(iii) For every $\varepsilon \geq 0$, we have $MBenson(S, E, C) \subseteq \bigcap_{\varepsilon > 0} (\varepsilon, e)$-MBenson$(S, E, C)$.*

Firstly, we derive the optimality conditions of $(\varepsilon, e)$-weak minimal efficient solution for $(\mathcal{SOP})$.

**Theorem 1.** *Let $\varepsilon > 0$, $e \in \text{int}(C)$ and $(t_0, q_0) \in V \times (-\text{int}(C))$. If $v_0 \in E$ is a $(\varepsilon, e)$-weak minimal solution of $(\mathcal{SOP})$, then*

$$\left(D''_{l-r}S_+(v_0, t_0, q_0)(N) + \varepsilon e\right) \cap (-\text{int}(C)) = \varnothing, \tag{7}$$

*where $N := \text{dom}\ D''_{l-r}S_+(v_0, t_0, q_0)$.*

**Proof.** Suppose that (7) dose not hold. Then, there exists $\bar{v} \in N$ and

$$\bar{p} \in D''_{l-r}S_+(v_0, t_0, q_0)(\bar{v})$$

such that

$$\bar{p} + \varepsilon e \in -\text{int}(C). \tag{8}$$

By the definition of the generalized second-order lower radial epiderivatives, for a sequence $\{m_n\}$ with $m_n = 1$, there exists $\{(v_n, p_n)\} \subseteq V \times P$ with $(v_n, p_n) \to (\bar{v}, \bar{p})$ such that

$$(v_0 + t_0 + v_n, p_0 + q_0 + p_n) \in \text{epi}\ S, \forall p_0 \in S(v_0) \text{ and } n \in \mathbb{N}.$$

Then, for every $p_0 \in S(v_0)$ and $n \in \mathbb{N}$, we get

$$p_0 + q_0 + p_n \in S(v_0 + t_0 + v_n) + C.$$

Therefore,

$$p_n \in S(v_0 + t_0 + v_n) - p_0 + C - q_0, \forall p_0 \in S(v_0), n \in \mathbb{N}.$$

In combination with $q_0 \in -C$, one gets

$$p_n \in S(v_0 + t_0 + v_n) \dot{-} S(v_0) - \{q_0\} + C$$
$$\subseteq S(v_0 + t_0 + v_n) \dot{-} S(v_0) + C, \forall n \in \mathbb{N}.$$

Since $\varepsilon > 0$ and $e \in \text{int}(C)$, one has

$$p_n + \varepsilon e \in S(v_0 + t_0 + v_n) \dot{-} S(v_0) + C + \varepsilon e, \forall n \in \mathbb{N}. \tag{9}$$

Obviously, $p_n + \varepsilon e \to \bar{p} + \varepsilon e$. It follows from (8) that there exists a natural number $N$ such that

$$p_n + \varepsilon e \in -\text{int}(C), \forall n > N.$$

In combination with (9), it follows from Proposition 1 $(d)$ that

$$(S(E) \dot{-} S(v_0) + C + \varepsilon e) \cap -\text{int}(C)) \neq \varnothing.$$

Therefore,

$$(S(E) \dot{-} S(v_0) + \varepsilon e) \cap -\text{int}(C)) \neq \varnothing.$$

which contradicts that $v_0$ is a $(\varepsilon, e)$-weak minimal solution of problem $(\mathcal{SOP})$. The proof is complete. $\square$

According to Proposition 6, we get that the following corollary.

**Corollary 1.** Let $\varepsilon > 0$, $e \in \text{int}(C)$ and $(t_0, q_0) \in V \times (-\text{int}(C))$. If $v_0 \in E$ is a $(\varepsilon, e)$-weak minimal solution of problem $(\mathcal{SOP})$, then

$$\left(D''_{l-r}S(v_0, t_0, q_0)(K) + \varepsilon e\right) \cap (-\text{int}(C)) = \varnothing,$$

where $K := \text{dom } D''_{l-r}S(v_0, t_0, q_0)$.

Now we give an example to explain Theorem 1.

**Example 3.** *Consider set optimization problem with* $V = \mathbb{R}$, $P = \mathbb{R}^2$, $E = V$, $C = \mathbb{R}^2_+$. *Let*

$$S(v) = \left\{(p_1, p_2) \in \mathbb{R}^2 \mid p_1 \geq v^2, p_1 + p_2 \geq v\right\}, v \in E.$$

*It is easy to check that* $v_0 = 1$ *is a* $(\varepsilon, e)$-*weak minimal solution of the problem* $(\mathcal{SOP})$. *Let* $\varepsilon = 1$ *and* $e = (1, 1)$. *Then, by directly calculating, we get*

$$D''_{l-r}S_+(v_0, t_0, q_0)(v) + \varepsilon e = \left\{(p_1, p_2) \in \mathbb{R}^2 \mid p_1 \geq 1, p_1 + p_2 \geq v + 1\right\}, v \in E.$$

*Then*

$$\left(D''_{l-r}S_+(v_0, t_0, q_0)(v) + \varepsilon e\right) \cap (-\text{int}(C)) = \varnothing, \forall v \in V.$$

**Remark 11.** *The condition of Theorem 1 is also a second-order necessary condition for m-weak minimal solution in [24].*

**Theorem 2.** Let $\varepsilon > 0$ and $e \in \text{int}(C)$. If there exists $(t_0, q_0) \in \{0\} \times C$ such that

$$D''_r S(v_0, t_0, q_0)((E - v_0) + \varepsilon e) \cap (-\text{int}(C)) = \varnothing, \tag{10}$$

then $v_0$ is a $(\varepsilon, e)$-weak minimal solution of $(\mathcal{SOP})$.

**Proof.** From (10), we derive

$$\left( D_r'' S(v_0, t_0, q_0)(E - v_0) + \varepsilon e + C \right) \cap (-\text{int}(C)) = \varnothing. \tag{11}$$

Suppose that (11) dose not hold. Then there exist $v \in E$, $p \in D_r'' S(v_0, t_0, q_0)(v - v_0)$ and $c \in C$ such that

$$p + \varepsilon e + c \in (-\text{int}(C)).$$

Since $\text{int}(C) + C \subseteq \text{int}(C)$, one has

$$p + \varepsilon e \in -\text{int}(C) - c \subseteq -\text{int}(C).$$

Obviously, $p + \varepsilon e \in D_r'' S(v_0, t_0, q_0)((v - v_0) + \varepsilon e)$. Therefore,

$$D_r'' S(v_0, t_0, q_0)((E - v_0) + \varepsilon e) \cap (-\text{int}(C)) \neq \varnothing.$$

which contradicts with (10). Hence (11) holds.

As $\varepsilon > 0$ and $e \in \text{int}(C)$, it follows from Proposition 7 that

$$S(E) \dot{-} S(v_0) + \varepsilon e \subseteq D_r'' S(v_0, t_0, q_0)(E - v_0) + \varepsilon a + K.$$

Combining with (11), we have

$$(S(E) \dot{-} S(v_0) + \varepsilon e) \cap (-\text{int}(C)) = \varnothing.$$

Therefore $v_0$ is a $(\varepsilon, e)$-weak minimal solution of $(\mathcal{SOP})$. The proof is complete. □

Now we give an example to show Theorem 2.

**Example 4.** *Consider* $(\mathcal{SOP})$ *with* $V = \mathbb{R}$, $P = \mathbb{R}^2$, $C = \mathbb{R}_+^2$ *and* $E = [1, 2]$. *Let*

$$\left\{ S(v) = (p_1, p_2) \in \mathbb{R} \mid p_1 \geq -\frac{1}{v}, p_2 \geq -\frac{1}{v}, -v \leq p_1 + p_2 \leq 2v \right\}, 1 \leq v \leq 2.$$

*Take* $v_0 = 1$. *Let* $\varepsilon = 1$ *and* $e = (1, 1)$. *It is easy to caculate that*

$$D_r'' S(v_0, t_0, q_0)((v - v_0) + \varepsilon e) = (\frac{1}{2}, \frac{1}{2}).$$

*Then*

$$D_r'' S(v_0, t_0, q_0)((E - v_0) + \varepsilon e) \cap (-\text{int}(C)) = \varnothing.$$

*Thus,* $v_0 = 1$ *is a* $(\varepsilon, e)$-*weak minimal solution.*

Since the $(\varepsilon, e)$-weak minimal solution is not always the $(\varepsilon, e)$-MBenson proper efficient solution for $(\mathcal{SOP})$, we next provide optimality conditions of the $(\varepsilon, e)$-MBenson proper efficient solution $(\mathcal{SOP})$.

**Theorem 3.** *Let* $\varepsilon > 0$, $e \in \text{int}(C)$, $v_0 \in E$ *and* $(t_0, q_0) \in E \times (-C)$. *If* $v_0$ *is a* $(\varepsilon, e)$-*MBenson proper efficient solution of problem* $(\mathcal{SOP})$, *then*

$$\left( D_{l-r}'' S_+(v_0, t_0, q_0)(N) + \varepsilon e \right) \cap (-C \setminus \{0\}) = \varnothing, \tag{12}$$

*where* $N := \text{dom } D_{l-r}'' S_+(v_0, t_0, q_0)$.

**Proof.** Suppose to the contrary that there exists some $\bar{v} \in N$ such that (12) does not hold. Then there exists some $\bar{p} \in P$ such that

$$\bar{p} \in D_{l-r}'' S_+(v_0, t_0, q_0)(\bar{v})$$

and

$$\bar{p} + \varepsilon e \in -C \setminus \{0\}. \tag{13}$$

By the definition of the generalized second-order lower radial epiderivatives, for a sequence $\{m_n\}$ with $m_n = 1$, there exists $\{(v_n, p_n)\} \subseteq V \times P$ with $(v_n, p_n) \to (\bar{v}, \bar{p})$ such that

$$(v_0 + t_0 + v_n, p_0 + q_0 + p_n) \in \text{epi } S, \forall p_0 \in S(v_0) \text{ and } n \in \mathbb{N}.$$

Then, for every $p_0 \in S(v_0)$ and $n \in \mathbb{N}$, we get

$$p_0 + q_0 + p_n \in S(v_0 + t_0 + v_n) + C.$$

Therefore,

$$p_n \in S(v_0 + t_0 + v_n) - p_0 + C - q_0, \forall p_0 \in S(v_0), n \in \mathbb{N}.$$

Combining with $q_0 \in -C$ and Proposition 1 $(b)$, one gets

$$\begin{aligned}
p_n &\in S(v_0 + t_0 + v_n) \dot{-} S(v_0) - \{q_0\} + C \\
&\subseteq S(E) \dot{-} S(v_0) - \{q_0\} + C \\
&\subseteq S(E) \dot{-} S(v_0) + C, \forall n \in \mathbb{N}.
\end{aligned}$$

Since $\varepsilon > 0$ and $e \in \text{int}(C)$, one has

$$p_n + \varepsilon e \in S(E) \dot{-} S(v_0) + C + \varepsilon e, \forall n \in \mathbb{N}.$$

So

$$\bar{p} + \varepsilon e \in \text{clcone}(S(E) \dot{-} S(v_0) + C + \varepsilon e).$$

Combining with (13), we get

$$\text{clcone}(S(E) \dot{-} S(v_0) + C + \varepsilon e) \cap (-C \setminus \{0\}) \neq \emptyset,$$

which contradicts that $v_0$ is a $(\varepsilon, e)$-MBenson proper efficient solution of problem $(\mathcal{SOP})$. The proof is complete. □

According to Proposition 6, we get that the following corollary.

**Corollary 2.** *Let $\varepsilon > 0$, $e \in \text{int}(C)$, $v_0 \in E$ and $(t_0, q_0) \in E \times (-C)$. If $v_0$ is a $(\varepsilon, e)$-MBenson proper efficient solution of problem $(\mathcal{SOP})$, then*

$$\left(D''_{l-r}S(v_0, t_0, q_0)(K) + \varepsilon e\right) \cap (-C \setminus \{0\}) = \emptyset,$$

*where $K := \text{dom } D''_{l-r}S(v_0, t_0, q_0)$.*

Now we provide an example to illustrate Theorem 3.

**Example 5.** *Consider set optimization problem with $V = \mathbb{R}$, $P = \mathbb{R}^2$, $E = V$, $C = \mathbb{R}^2_+$. Let*

$$S(v) = \left\{(p_1, p_2) \in \mathbb{R}^2 \mid p_1 \geq 0, p_2 \geq v^2\right\}, v \in E.$$

It is easy to check that $v_0 = 1$ is a $(\varepsilon, e)$-MBenson proper efficient solution of the problem $(\mathcal{SOP})$. Let $\varepsilon = 1$ and $e = (1, 1)$. Then, by directly calculating, we get

$$D''_{l-r}S_+(v_0, t_0, q_0)(v) + \varepsilon e = \left\{(p_1, p_2) \in \mathbb{R}^2 \mid p_1 \geq 0, v \geq p_2 \geq v^2\right\}, v \in E.$$

Then

$$\left(D''_{l-r}S_+(v_0, t_0, q_0)(v) + \varepsilon e\right) \cap (-C \setminus \{0\}) = \emptyset, \forall v \in E.$$

**Remark 12.** *If $\varepsilon = 0$ and $e \in C$ in Theorem 3, it follows from Remark 10 (i) that $v_0$ becomes a MBenson proper efficient solution of problem $(\mathcal{SOP})$, and (12) becomes the necessary condition for MBenson proper efficient solution.*

**Remark 13.** *As $MBenson(F, E, C) \subseteq (\varepsilon, e)$-$MBenson(F, E, C)$ from Theorem 3, so the condition of Theorem 3 is also a second-order necessary condition for MBenson proper efficient solution.*

**Remark 14.** *If the condition of $q \in -C$ is not satisfied in Theorem 3, then Theorem 3 may not hold. The following example explains the case.*

**Example 6.** *Consider Example 5. Then $v_0 = 1$ is a $(\varepsilon, e)$-MBenson proper efficient solution of the problem $(\mathcal{SOP})$. Take $q = (1, 0) \notin -C$ and*

$$D''_{l-r}S_+(v_0, t_0, q_0)(v) + \varepsilon e = \left\{ (p_1, p_2) \in \mathbb{R}^2 \mid p_1 \in \mathbb{R}, v \geq p_2 \geq v^2 \right\}.$$

*Thus,*

$$\left( D''_{l-r}S_+(v_0, t_0, q_0)(v) + \varepsilon e \right) \cap (-C \setminus \{0\}) \neq \emptyset, \forall v \in E.$$

**Theorem 4.** *Let $v_0 \in E$, $\varepsilon > 0$, $e \in \text{int}(C)$ and $(t_0, q_0) \in \{0\} \times K$. If*

$$\text{clcone}\left( D''_r S(v_0, t_0, q_0)(E - v_0) + C + \varepsilon e \right) \cap (-C \setminus \{0\}) = \emptyset, \tag{14}$$

*then $v_0$ is a $(\varepsilon, e)$-MBenson proper efficient solution of problem $(\mathcal{SOP})$.*

**Proof.** By Proposition 7, we have

$$S(v) \dot{-} S(v_0) - \{q_0\} \subseteq D''_r S(v_0, t_0, q_0)(v - v_0 - t_0) + C, \quad \forall v \in E.$$

Since $(t_0, q_0) \in \{0\} \times C$, one has

$$S(v) \dot{-} S(v_0) \subseteq D''_r S(v_0, t_0, q_0)(v - v_0) + C, \quad \forall v \in E.$$

Then, from Proposition 1 (*d*), we have

$$S(E) \dot{-} S(v_0) \subseteq D''_r S(v_0, t_0, q_0)(E - v_0) + C.$$

Therefore, combining with $C + C = C$, we get

$$\text{clcone}(S(E) \dot{-} S(v_0) + C + \varepsilon e) \subseteq \text{clcone}\left( D''_r S(v_0, t_0, q_0)(E - v_0) + C + \varepsilon e \right).$$

Then it follows from (14) that

$$\text{clcone}(S(E) \dot{-} S(v_0) + C + \varepsilon e) \cap (-C \setminus \{0\}) = \emptyset,$$

which implies that $v_0$ is a $(\varepsilon, e)$-MBenson proper efficient solution of problem $(\mathcal{SOP})$. This completes the proof. $\square$

Now we give an example to illustrate Theorem 4.

**Example 7.** *Consider $(\mathcal{SOP})$ with $V = \mathbb{R}$, $P = \mathbb{R}^2$, $C = \mathbb{R}^2_+$ and $E = [1, 2]$. Let*

$$\{S(v) = (p_1, p_2) \in \mathbb{R} \mid p_1 \geq v, P_1 + P_2 \geq v\}, 1 \leq v \leq 2.$$

*Take $v_0 = 1$. Let $\varepsilon = 1$ and $e = (1, 1)$, then by calculating, we get*

$$D''_r S(v_0, t_0, q_0)((v - v_0) + \varepsilon e) = \left\{ (p_1, p_2) \in \mathbb{R}^2 \mid p_1 \geq 0, p_1 + p_2 = v \right\}, v \in E.$$

*Thus,*

$$\text{clcone}\big(D_r'' S(v_0, t_0, q_0)(E - v_0) + C + \varepsilon e\big) \cap (-C \setminus \{0\}) = \varnothing,$$

*which implies that $v_0 = 1$ is a $(\varepsilon, e)$-MBenson proper efficient solution.*

**Remark 15.** *if we replace the $(\varepsilon, e)$-MBenson proper efficient solution with the MBenson proper efficient solution in Theorems 3 and 4, then the corresponding conclusions are still valid.*

### 5. Conclusions

In this paper, we firstly propose the notion of the generalized second-order lower radial epiderivatives and discuss some properties about it. We also extend a few crucial properties of generalized second-order radial derivatives. Finally, we establish the necessary and sufficient optimality conditions of approximate Benson proper efficient solutions and approximate weakly minimal solutions for the set optimization problem under the unconstrained condition.

It is significant to emphasize that no prior research has been done on the optimality conditions for approximation solutions with the Minkowski difference of set optimization problems, which is the subject of this paper. It would be great to investigate these ideas by using the new derivative.

**Author Contributions:** Conceptualization, Y.Z. and Q.W.; methodology Y.Z. and Q.W.; writing-original draft Y.Z.; writing-review and editing Y.Z. and Q.W. All authors have read and agreed to the published version of the manuscript.

**Funding:** This research was partially supported by the National Natural Science Foundation of China (No.11971078), the Group Building Project for Scientifc Innovation for Universities in Chongqing (CXQT21021), Science and technology research project of Chongqing Municipality Education Commission (KJZD-K202300708), Joint Training Base Construction Project for Graduate Students in Chongqing (JDLHPYJD2021016), and the Graduate Student Science and Technology Innovation Project of Chongqing Jiaotong University (2022ST003).

**Data Availability Statement:** Not applicable.

**Conflicts of Interest:** The author declares no conflict of interest.

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
