# Peer review of "Optimality Conditions for Approximate Solutions of Set Optimization Problems with the Minkowski Difference"

_axioms, doi:10.3390/axioms12101001_

Round 1

Reviewer 1 Report

An article on optimality conditions for approximate solution of set optimization problems with Minkowski difference is presented for consideration.

The Minkowski difference has an interesting property: if two figures overlap or intersect, the resulting Minkowski difference will contain the origin. And this is the basis of the GJK algorithm, which is suitable for determining the distance between two convex sets.

In the work, the authors established several important properties of the Minkowski difference for sets, and also introduced a generalized lower radial epiderivative of the second order for multivalued mappings with respect to the Minkowski difference. Based on this, necessary and sufficient conditions for the optimality of Benson's approximate eigenefficient solutions were established.

In general, the work is relevant, expands and complements questions on the application of set optimization problems with Minkowski differences, and deserves to be published after minor comments have been eliminated:

1. The introduction can be slightly reworked; it is recommended to use a link to a maximum of 3 sources at the same time, for example, [1-3], [16,17,23], but not [4–8,15,17,30].

2. In the list of references, more than 70% of sources are older than 5 years, and 50% are older than 10 years. I would recommend using newer sources.

3. In the conclusion, it is not allowed to refer to sources from the list of references, since the conclusion is a summary of the results of this article. This needs to be fixed.

Reviewer 2 Report

English is ok. However, minor spell and typos to be checked.

Reviewer 3 Report

Report on the paper

Optimality conditions for approximate solutions of set optimization problems with the Minkowski difference

by Yuhe Zhang and Qilin Wang

The authors' aim is to 'study the optimality conditions for set optimization problems with set criterion'.

They use a variant of Minkowski difference, used previousely in the litterature to have a sort of a partial inversion of Minkowski sum.

The obtained results are interesting, and some examples are inserted in the text.

Authors must be careful in using results and citing them; for example Lemma 2 of the authors is mentioned as a marginal result in [27], but the authors do not mention this. It is not a central result of the work, but we believe that mathematical rigor requires it.

Overall, however, we believe that the work is valuable and deserves to be published, after a minor revision.
